# The Potential of *Pediococcus acidilactici* Cell-Free Supernatant as a Preservative in Food Packaging Materials

**DOI:** 10.3390/foods13050644

**Published:** 2024-02-21

**Authors:** Katherine Kho, Adinda Darwanti Kadar, Mario Donald Bani, Ihsan Tria Pramanda, Leon Martin, Matthew Chrisdianto, Ferren Pratama, Putu Virgina Partha Devanthi

**Affiliations:** Department of Biotechnology, Faculty of Life Sciences, Indonesia International Institute for Life Sciences, Pulomas Barat Kavling 88, Jakarta 13210, Indonesia; adinda.kadar@i3l.ac.id (A.D.K.); mario.bani@i3l.ac.id (M.D.B.); ihsan.pramanda@i3l.ac.id (I.T.P.); leon.martin@alumni.i3l.ac.id (L.M.); matthew.chrisdianto@alumni.i3l.ac.id (M.C.); ferren.pratama@alumni.i3l.ac.id (F.P.); putu.devanti@i3l.ac.id (P.V.P.D.)

**Keywords:** *Pediococcus acidilactici*, cell-free supernatant, postbiotics, organic acids, food packaging, natural preservatives

## Abstract

This study delves into the production and antimicrobial characteristics of cell-free supernatants from *Pediococcus acidilactici* (CFSs-Pa). Antimicrobial activity was initially observed in CFS-Pa harvested after 12 h of incubation and increased up to the late stationary phase at 48 h. The increase in antimicrobial activity did not align with total protein content, pointing to other factors linked to the accumulation of organic acids, particularly lactic acid. The SDS-PAGE analysis also indicated that the expected proteinaceous compound (pediocin) was not observed in CFS-Pa. Further investigations suggested that the antimicrobial properties of CFS-Pa were exclusively due to organic acids. The MIC values confirmed potent antimicrobial activity, particularly at a 10% dilution of CFS-Pa in MRS broth. The time–kill assays demonstrated bactericidal activity against EHEC, *Listeria monocytogenes,* and *Staphylococcus aureus* by 12 h, 18 h, and 24 h using a 10% dilution of CFS-Pa. Additionally, CFS-Pa exhibited dose-dependent antioxidant activity, requiring a 70% (*v*/*v*) concentration to inhibit DPPH scavenging activity by 50%. All the experimental results suggested potential applications of CFS-Pa in food preservation. An attempt to incorporate CFS-Pa into bacterial cellulose (BC) for edible food packaging demonstrated promising antimicrobial results, particularly against *L. monocytogenes* and *S. aureus*, with room for optimization.

## 1. Introduction

The use of synthetic chemical preservatives (e.g., benzoic acid, sodium benzoate, parabens, nitrite, and sodium sulfite) in food products has long been an effective strategy to extend shelf life, prevent spoilage, and improve food safety [1]. These preservatives work by inhibiting the growth of undesirable microorganisms and by slowing down oxidative processes that can lead to food deterioration. However, concerns about potential health risks [1], allergic reactions [2], and the emergence of antibiotic resistance [3] have shifted the attention of both the food industry and consumers toward natural alternatives. Plant extracts [4] and essential oils [5] are among the popular natural alternatives, providing antioxidant and antimicrobial properties. Besides plant extracts, antimicrobial compounds derived from microbes stand out as a promising natural preservative. Microbes produce a vast array of secondary metabolites with diverse chemical structures, resulting in a broader spectrum of antimicrobial activity [6]. The high specificity of microbial compounds allows for targeted action against pathogens while minimizing harm to beneficial microorganisms [7]. Furthermore, microbial production is scalable through fermentation processes, providing cost-effective and rapid production compared to plants, and it is independent of seasonal variability. Moreover, microbes are relatively easier to genetically modify, allowing for further optimization.

Lactic acid bacteria (LAB), such as Lactobacillus and Pediococcus, have played a significant role in the preservation of many food products, such as dairy products, fermented vegetables, and meats. LAB are generally regarded as safe (GRAS) and have long been utilized in the fermentation of various foods, contributing not only to flavor development and textural improvement but also to food safety and preservation. Their remarkable antimicrobial properties lie in their ability to produce lactic acid as a metabolic byproduct, creating an acidic environment that inhibits the growth of spoilage and pathogenic microorganisms. Moreover, LAB are known for synthesizing bacteriocins, which are small antimicrobial peptides with specificity against various bacteria [7,8]. Beyond their preservative capabilities, LAB are recognized for their probiotic properties, with increasing evidence supporting their role in promoting gut health.

Despite its antimicrobial benefits, the incorporation of live bacterial cells into food products can be unfavorable in certain cases, as they may interfere with product characteristics or sensory qualities, especially when their metabolic byproducts are not desirable. In such cases, the use of cell-free supernatant (CFS) becomes an attractive alternative. CFS refers to the residual nutrients and metabolites produced by microorganisms during fermentation, including antimicrobial substances (e.g., organic acids, hydrogen peroxide, antifungal peptides, and bacteriocins) [9], and antioxidants (e.g., phenolic and flavonoid compounds) [10]. The CFS of LAB has previously been reported for its antimicrobial activity against Gram-positive and Gram-negative bacteria. However, its effectiveness depends on the species. For example, *L. plantarum* has shown the highest antimicrobial activity among the other 16 LAB when tested against *Escherichia coli*, *Staphylococcus aureus*, *Shigella sonnei*, *Pseudomonas fluorescens*, *Salmonella typhimurium*, and *Listeria monocytogenes* [11].

*Pediococcus acidilactici* is one of the LAB that has captured considerable interest in recent years due to its exceptional antimicrobial properties. A study conducted by Fugaban et al. [8] showed that the CFS obtained from *P. acidilactici* ST3522BG and *P. pentosaceus* ST3633BG can effectively inhibit the growth of *L. monocytogenes*, one of the most significant foodborne pathogens responsible for listeriosis. Additionally, another study by Kim and Kang [12] found that CFS derived from *P. acidilactici* HW01 exhibits antifungal properties against *Candida albicans* and its biofilm formation. More recently, Liu et al. [13] found that CFS derived from *P. acidilactici* LWX 401 exhibited a broad-spectrum antimicrobial activity against multidrug-resistant Gram-negative EHEC and *S. aureus*. The metabolites that play a major role in the antimicrobial activity of the CFS from *P. acidilactici* (CFS-Pa) are bacteriocin-like peptides, mainly pediocin, and organic acids. Pediocin is a small antimicrobial peptide, generally less than 5 kDa, that falls into the class IIa bacteriocins. However, the production of pediocin appears to be strain-dependent in *P. acidilactici* since not all strains of this bacterial species can produce this antimicrobial peptide [8]. Pediocin itself has been known to be able to inhibit the growth of many food spoilage bacteria and pathogens, including *Listeria monocytogenes*, *Staphylococcus aureus, Escherichia coli* O157:H7 (EHEC), and *Salmonella typhi* [8,14,15]. It is generally recognized that, besides pediocin, organic acids also play a major role in this inhibition [7,12,16,17]. In fact, a recent metabolomic study on the CFS of *P. acidilactici* LWX 401 demonstrated that organic acids, along with lipid molecules, accounted for 49.03% of the total upregulated metabolites identified [13]. The production of organic acids can lower the pH of the surrounding environment, rendering it unsuitable for the growth of undesirable bacteria. In addition, *P. acidilactici* also produces hydrogen peroxide, which plays a role in this bacterium’s antimicrobial activity [18]. In addition, one class of the metabolites detected, alkaloids, could potentially act as antioxidant compounds [13,19]. Nevertheless, the antioxidant properties of CFS-Pa have not been explored.

It is worth noting that the concentration of individual antimicrobial compounds in the CFS often falls below the minimum inhibitory concentration (MIC). This implies that the antimicrobial activity of LAB is not solely due to one compound but rather the synergistic effects of multiple compounds in the CFS cocktail [20]. The composition and concentration of these metabolites can vary based on several factors, which affect the overall efficacy of CFS as a natural preservative. One crucial factor influencing the profile of these antimicrobial substances is the incubation time during bacterial fermentation. Guerra, Bernardez, and Castro reported the production of different metabolites at different growth phases of *P. acidilactici* NRRL B-5627 grown in whey media with 2% yeast extract [21]. Lactic acid and ethanol concentration peaked before 15 h, while pediocin production peaked after 30 h.

Although previous reports have shown efforts in improving and characterizing the antimicrobial activity of *P. acidilactici*’s CFS, the optimal incubation period for harvesting *P. acidilactici* and utilizing its CFS as an antimicrobial agent in food preservation remains an open discussion. Further research is required to determine which incubation period yields the highest antimicrobial activity from CFS-Pa. This information is crucial for maximizing the potential of *P. acidilactici* in food preservation and ensuring its efficacy as an antimicrobial agent.

Therefore, this study aimed to determine the optimal incubation period for *P. acidilactici*, resulting in the production of CFS-Pa with the highest antimicrobial activity. The effectiveness of CFS-Pa was evaluated against foodborne pathogens, including *Staphylococcus aureus* (*S. aureus*), *Listeria monocytogenes* (*L. monocytogenes*), and Enterohaemorrhagic *Escherichia coli* EHEC. Furthermore, the CFS-Pa exhibiting the highest antimicrobial activity was subjected to further investigation to determine whether this activity was attributed to organic acids and/or bacteriocin-like compounds. Additionally, the antioxidant activity of CFS-Pa was examined by measuring its ability to scavenge free radicals (DPPH). Finally, this study also evaluated the possible application of CFS-Pa in conferring antimicrobial properties to bacterial cellulose (BC) produced by *Komagataeibacter intermedius* (*K. intermedius*), as a biodegradable material with excellent features for novel edible coatings and food packaging [22]. This approach has significant potential to improve the development of sustainable and functional food packaging materials.

## 2. Materials and Methods

### 2.1. Materials and Culture

The *P. acidilactici* was obtained from the Indonesia International Institute for Life Sciences (i3L) culture collection. The materials used for the microbiological growth media were de Man–Rogosa–Sharpe (MRS) broth (Merck, Darmstadt, Germany) and MRS agar (Merck, Darmstadt, Germany). The bacteria were cultured on MRS agar at 30 °C for 48 h and stored at 4 °C. *K. intermedius* was also obtained from the i3L culture collection with the code (13.KBC.HS) and was maintained on MRS agar at 30 °C and stored at 4 °C.

The pathogens used in this study were *S. aureus,* Enterohaemorrhagic *Escherichia coli* (EHEC), and *L. monocytogenes*. Both *S. aureus* and EHEC were obtained from the i3L culture collection, while *L. monocytogenes* was purchased from PT. Alpha Science Innolab, Jakarta, Indonesia. These pathogens were cultured in non-selective nutrient broth media (Merck, Darmstadt, Germany) at 37 °C for approximately 24 h. Then, the pathogens were diluted into 6 Log CFU/mL in Mueller–Hinton broth (Himedia Laboratories Pvt Ltd., Mumbai, India), based on the Clinical and Laboratory Standards Institute (CLSI) standard for antimicrobial testing. *S. aureus*, EHEC, and *L. monocytogenes* were enumerated using selective media, including mannitol salt agar (MSA) (Himedia Laboratories Pvt Ltd., Mumbai, India), sorbitol MacConkey agar (SMAC) (Himedia Laboratories Pvt Ltd., Mumbai, India), and polymyxin–acriflavine–lithium chloride–ceftazidime–esculin–mannitol (PALCAM) Listeria Identification Agar (Himedia Laboratories Pvt Ltd., Mumbai, India), respectively. The viable cell count used NaCl as the diluent for serial dilution and the antimicrobial testing used Mueller–Hinton broth (MHB) media along with chloramphenicol as the antimicrobial reference (all three materials were purchased from Himedia Laboratories Pvt Ltd., Mumbai, India).

The CFS-Pa characterization utilized the Bicinchoninic (BCA) Protein Quantification Kit E112 (Vazyme Biotech, Nanjing, China) for total protein quantification, Megazyme L-Lactic Acid K-LATE Kit (Megazyme Ltd., Bray, Republic of Ireland) for lactic acid measurement, pepsin (Sigma-Aldrich, Burlington, MA, USA) and catalase (Merck, Darmstadt, Germany) for both pepsin and catalase digestion assays, 2,2-diphenyl-1-picrylhydrazyl (DPPH, Sigma-Aldrich, Burlington, MA, USA) for the antioxidant activity assay. Lactic acid (Sigma-Aldrich, Burlington, MA, USA) was also used as a reference for several characterization assays.

The SDS-PAGE components consisted of running buffer, sample buffer, staining solution, destaining solution, and cast gel reagents. The SDS-PAGE running buffer consisted of 0.025 M Tris, 0.192 M glycine, and SDS. The sample buffer consisted of Laemmli buffer and β-mercaptoethanol. The staining solution consisted of Coomassie brilliant blue G-250, acetic acid (both from Merck, Darmstadt, Germany), and methanol. The destaining solution also consisted of methanol and acetic acid. The reagents for the casting gel consisted of 1.5M Tris-HCl, SDS, acrylamide, APS, TEMED, and isopropanol. The SDS-PAGE used Spectra™ Multicolor Broad Range Protein Ladder (10 to 260 kDa, Thermo Fisher Scientific, Massachusetts, USA) for the protein ladder and also bovine serum albumin (Vazyme Biotech, Nanjing, China) as a reference.

### 2.2. Cell-Free Supernatant (CFS) Production

The *P. acidilactici* culture was reactivated by inoculating 10% (*v*/*v*) of an active culture into 100 mL MRS broth in separate Erlenmeyer flasks. Multiple flasks were utilized to represent distinct time points (0, 6, 12, 24, 36, 48 h), resulting in a total of 18 flasks, including biological triplicates. After a certain number of hours of incubation, the cultures were separated into 2 Falcon tubes (50 mL) and centrifuged at 5000 rpm and 4 °C for 36 min. The pellets were discarded while the supernatants were filtered using syringe filters 0.22 μm (Wuxi NEST Biotechnology Co., Ltd., WuXi, China; cat. no. 331011). The resulting CFSs-Pa were divided into two batches, where one was stored at 4 °C while the other was stored at −70 °C for the next experiment. In addition, a sample was also collected every 3 h and subjected to the Miles and Misra viable cell counting method described above using MRS agar for the growth curve.

### 2.3. Viable Cell Count

The viable cell count was performed using the Miles and Misra method with adjustments [23]. Briefly, serial dilution was performed by transferring 100 μL of the sample into a series of microcentrifuge tubes containing 900 μL of 0.9% NaCl (Himedia Laboratories Pvt Ltd., Mumbai, India), yielding 10-fold serial dilutions. Then, 10 μL of each dilution factor was pipetted on MRS agar in triplicate, followed by incubation. The dilution factor consisting of 3–30 colonies was used in Equation (1) to calculate the viability.
(1)$$Log\;CFU/mL=log10(\frac{number\;of\;colonies\;\times \;dilution\;factor}{0.01mL})$$

### 2.4. Antimicrobial Activity Testing with Agar Well Diffusion

The agar well diffusion testing was adapted from [11] with several adaptations. The Mueller–Hinton agar (Himedia Laboratories Pvt Ltd., Mumbai, India) was prepared at 10 mL volume, where 0.1 mL of pathogen cultures was spread using a cell spreader. Three wells were punched in the plate using sterile 1 mL micropipette tips. Each well was filled with 50 μL of CFS-Pa sample. The plates were incubated for 24 h at 37 °C, and the zone of inhibition (mm) was measured using a digital caliper.

### 2.5. CFS-Pa Characterization

#### 2.5.1. Total Protein Quantification Using Bicinchoninic Acid (BCA) Assay

The protein concentration of the CFS-Pa was measured using a Bicinchoninic (BCA) Protein Quantification Kit E112 (Vazyme Biotech, Nanjing, China) following the manufacturer’s standard protocol. Initially, the BCA working solution was prepared by mixing reagent A and reagent B at a 50:1 ratio. The standard curve was generated by measuring seven concentrations of BCA working solutions ranging from 0 to 10 μL diluted in water. The BCA working solutions at 0 μL and 10 μL served as blank and reference controls, respectively. For the BCA assay itself, all CFS-Pa samples were diluted 10-fold initially. Then, 10 μL of diluted CFS-Pa samples from each time point was mixed with 100 μL of BCA working solution in a 96-well plate before being mixed briefly. The samples were then incubated at 37 °C for 20–30 min. The absorbance of both the standard curve and CFS-Pa samples was measured at 562 nm (A562) using a UV spectrophotometer (Infinite M200 NanoQuant, TECAN, Männedorf, Switzerland).

#### 2.5.2. Lactic Acid Profile Measurement

The lactic acid concentration of the CFS-Pa was measured using a Megazyme L-Lactic Acid K-LATE Kit (Megazyme Ltd., Bray, Ireland) following the manufacturer’s standard protocol. Initially, the standard curve was generated by measuring five concentrations of the standard solution ranging from 0 to 0.15 mg/mL, all diluted in 150 μL distilled water. The standard working solutions at 0 mg/mL and 0.15 mg/mL were the blank and reference controls, respectively.

For the lactic acid assay, 10 μL CFS-Pa samples from each time point were mixed with 150 μL of distilled water in a 96-well plate. The samples were added with 3 reagents: 50 μL of pH 10 buffer, 10 μL of NAD+/PVP solution, and 2 μL of D-GPT suspension. The samples were mixed briefly and then underwent absorbance measurement at 340 nm (A340) after 3 min using a UV spectrophotometer (Infinite M200 NanoQuant, TECAN, Männedorf, Switzerland). Then, 2 μL of L-LDH suspension was added to the samples to start the reaction. The absorbance was then measured at 5 min intervals for 15 min.

#### 2.5.3. pH Adjustment/Neutralization

To identify the nature of the antimicrobial metabolites in the CFS, their pH was neutralized to determine if the antimicrobial activity was due to organic acids. The pH of 5 mL of CFS-Pa was determined by electrode immersion with a pH meter (Starter ST3100, Ohaus, NJ, USA). Then, 4 mL of each CFS-Pa was taken, adjusted to pH 7 with a 0.5 M NaOH solution, and sterilized. Then, the neutralized CFS (CFS-N) was subjected to agar well diffusion. In addition, 100 µL of each sample was transferred to a 96-well plate alongside 100 µL of pathogens (6 Log CFU/mL). The 96-well plate was incubated for 24 h at 37 °C. The viability of pathogens after 24 h was assessed using the Miles and Misra plate count assay on selective agar specific to each pathogen.

#### 2.5.4. Pepsin Digestion Assay

The pepsin digestion assay was adapted from [13] with slight modification. The CFS-Pa was treated by proteolytic enzymes to determine if the antimicrobial activity was due to antimicrobial peptides. The CFS-Pa was treated individually with pepsin (0.5 mg/mL) in a 1:1 ratio for 2 h at 37 °C. Subsequently, the proteolytic enzymes were inactivated by heating the sample in a 95 °C water bath for 15 min. Then, 100 µL of each sample was transferred to a 96-well plate alongside 100 µL of pathogens (6 Log CFU/mL). The 96-well plate was incubated for 24 h at 37 °C. The viability of pathogens after 24 h was assessed using the Miles and Misra plate count assay on selective agar specific to each pathogen.

#### 2.5.5. Catalase Activity Assay

To identify the presence of hydrogen peroxides in the CFS, the CFS-Pa was treated with catalase to determine if the antimicrobial activity was due to the presence of hydrogen peroxide. The method was adapted from Kim and Kang [12]. The CFS-Pa was treated with catalase (1 mg/mL) in a 1:1 ratio for 2 h at 37 °C. Subsequently, the proteolytic enzymes were inactivated by heating the sample in a 95 °C water bath for 15 min. Then, 100 µL of each sample was transferred to a 96-well plate alongside 100 µL of pathogens (6 Log CFU/mL). The 96-well plate was incubated for 24 h at 37 °C. The viability of pathogens after 24 h was assessed using the Miles and Misra plate count assay on selective agar specific to each pathogen.

#### 2.5.6. SDS-PAGE Analysis

The presence of putative bacteriocin was checked by running the CFS and the precipitated sample in SDS-PAGE [24]. Here, Tris-glycine sodium dodecyl sulfate–polyacrylamide gel electrophoresis was used, employing a vertical slab gel apparatus (MiniPROTEAN Tetra Cell, BioRad, Hercules, CA, USA) with 4% stacking gel and 12% separating gel. SDS-PAGE was run at 100 V for 1.5 h along with the Spectra™ Multicolor Broad Range Protein Ladder (10 to 260 kDa, Thermo Fisher Scientific). Upon completion, the gel was stained with Coomassie brilliant blue G-250 (Merck, Darmstadt, Germany) and destained using a methanol and acetic acid solution with a 3:1 ratio. The proteins observed below ~10 kDa of the ladder were putatively identified as bacteriocin.

#### 2.5.7. Minimal Inhibitory Concentration (MIC) Test

The MIC test to determine the lowest concentration of an antimicrobial agent (like an antibiotic or antifungal) required to inhibit visible growth was adapted from [13] with modifications. The CFS-Pa samples were subjected to several concentrations (100%, 80%, 60%, 40%, 20%, and 10%) by dilution in MRS broth, with 100 µL of each sample transferred to a 96-well plate alongside 100 µL of pathogens (6 Log CFU/mL). This procedure resulted in lower CFS-Pa concentration after being diluted by the pathogens used as the final concentration for the test. The 96-well plate was incubated for 24 h at 37 °C. The viability of pathogens after 24 h was assessed using the Miles and Misra plate count assay on selective agar specific to each pathogen.

#### 2.5.8. Time–Kill Assay

The time–kill assay for antimicrobial activity was adapted from Prabhurajeshwar and Chandrakanth [25] and Kim and Kang [12] with several adjustments. The CFS-Pa and chloramphenicol (as an antimicrobial reference, diluted in Mueller–Hinton broth; Himedia, Mumbai, India) concentration was prepared according to the MIC test results for each pathogen. Then, 100 µL from each sample was transferred into a 96-well plate alongside 100 µL of pathogens, followed by incubation at 37 °C. The sample collection was performed at 6 h intervals for 24 h, and the viable pathogens were enumerated in a selective agar specific to each pathogen.

#### 2.5.9. Antioxidant Activity Assay

The antioxidant activity of the CFS-Pa was evaluated by using the 2,2-diphenyl-1-picrylhydrazyl (DPPH, Sigma-Aldrich) free radical scavenging assay as described by De Marco et al. [26], with some modifications. CFS-Pa was diluted in ethanol at different concentrations (20, 40, 60, 80, and 100% *v*/*v*). Then, 100 µL of the sample was added to 100 µL ethanol solution of DPPH (0.15 mM) in a 96-well microtiter plate. After 30 min of reaction at room temperature in the dark, the absorbance of each solution was read at 517 nm by a UV spectrophotometer (Infinite M200 NanoQuant, TECAN, Männedorf, Switzerland). The mixture of ethanol and sample was used as a blank. The control solution was prepared by mixing ethanol and DPPH radical solution. The antioxidant scavenging activity was calculated using Equation (2).
(2)$$DPPH\;Scavenging\;Activity\;(\%)=100-\left[(A517\;Sample-A517\;Blank)\times \frac{100}{A517\;Control}\right]$$

A517 sample is the absorbance of the sample after 30 min of reaction. A high percentage of scavenging activity indicates a higher amount of antioxidants present to scavenge free radicals.

### 2.6. Production and Antimicrobial Activity of CFS-Pa-Loaded Bacterial Cellulose

#### 2.6.1. Bacterial Cellulose Production by *Komagataeibacter intermedius*

Prior to bacterial cellulose (BC) cultivation, *K. intermedius* was grown in MRSB for 24 h at 30 °C and then subcultured again into fresh MRSB. After 5 days of incubation, the concentration of the active *K. intermedius* culture was adjusted to 6 log CFU/mL and was used as inoculum for BC production. BC was produced using 24-well plates where each well consisted of 1.8 mL fresh MRSB and 0.2 mL of *K. intermedius* culture. The plates were then incubated at 30 °C for 10 days in static conditions. Afterward, the BC was harvested and rinsed in distilled water to remove leftover media and treated with 1 M Na_2_CO_3_ at 80 °C until the BC became clear. Sodium carbonate was used for alkali treatment as it was more affordable, food grade, and less harsh compared to sodium hydroxide [27]. The BC was then neutralized with 25% acetic acid and then rinsed with distilled water. The treated BC was then sterilized at 121 °C for 15 min and then stored at 4 °C until further testing.

#### 2.6.2. CFS-Pa-Loaded Bacterial Cellulose Preparation and Antimicrobial Activity Testing

CFS-Pa was loaded into BC in 24-well plates where each well contained 1 mL of CFS-Pa along with 1 piece of BC (~0.8 mm thickness). Previous research has shown that the optimal loading time for BC ranged from 4 to 8 h [28], hence the BC was loaded with CFS-Pa for 8 h at 30 °C with agitation at 150 rpm. Afterward, the CFS-Pa-loaded BC samples were carefully scooped out and gently tapped against the wall of the 24-well plate to remove excess CFS-Pa. Finally, they were placed on MHA that had been spread with either *S. aureus*, EHEC, or *L. monocytogenes* (~6 log CFU/mL). The plates were then incubated at 37 °C for 24 h and the zone of inhibition was measured using a vernier caliper. The test was performed in triplicates against each pathogen.

### 2.7. Data Analysis

All the experiments were performed in triplicates. Data from the experiments were analyzed using GraphPad Prism version 8.0.0 for Windows, GraphPad Software, San Diego, CA, USA, www.graphpad.com, accessed on 1 November 2023. The effects were considered significant at *p* < 0.05. The results are presented as the mean ± standard deviation (SD). Statistical analysis on all tests was performed by one-way analysis of variance (ANOVA), except for the time-kill assay, which used two-way ANOVA. Tukey’s honestly significant difference test (*p* < 0.05) was conducted as a posteriori contrast after rejecting the null hypothesis.

## 3. Results and Discussion

### 3.1. The Optimal Incubation Period for the Production of CFS-Pa with the Highest Antimicrobial Activity

In this study, CFS-Pa was produced by growing *P. acidilactici* in MRS broth at 37 °C for 48 h without agitation. These conditions have been reported as optimal for the production of pediocin by LAB [29]. As shown in Figure 1a, *P. acidilactici* grew exponentially during the first 12 h of incubation without any observed lag phase. Afterwards, the number of cells increased slowly, indicating a late exponential phase between 12 and 18 h, followed by a stationary phase until the end of the incubation period. The antimicrobial activity of CFS-Pa was observed only after 12 h, during the late exponential phase (Figure 1b). However, it is worth noting that the antimicrobial activity observed at this stage (12 h) was limited to *S. aureus* and *L. monocytogenes* (Gram-positive bacteria), with the zones of inhibition measuring 3.6 ± 0.1 mm and 4.6 ± 0.8 mm, respectively. This raises a question as to whether the antimicrobial activity observed at this stage was due to pediocin or other proteinaceous substances since pediocin is known to be more effective against Gram-positive bacteria. Therefore, we also measured the total protein content in CFS-Pa during the incubation period to determine if it correlated with the antimicrobial activity of CFS-Pa against pathogens. The results showed a significant increase in the total protein content of CFS-Pa between 0 and 6 h (Figure 1c); however, no antimicrobial activity was observed during this period for any of the test pathogens. From 6 h onwards, the protein content remained relatively constant, while the increase in antimicrobial activity, particularly against Gram-positive bacteria, was observed. These findings could indicate that the increase in antimicrobial activity does not correlate with total protein content in the CFS-Pa. We also performed an SDS-PAGE analysis to further confirm the presence of any antimicrobial peptides, including pediocin (Appendix A). The results revealed no bands corresponding to pediocin or other antimicrobial peptides, which typically have a molecular size of around 5 kDa [30,31,32]. There were protein bands detected in all CFS samples after fermentation (T6-T48). However, a band of the same size was also detected in the CFS sample T0. This suggests that the bands may be associated with other protein content that could have come from the MRS media. Nevertheless, the band intensity becomes thicker over time (T6–T48), suggesting that these proteins might have also been secreted. The proteins might have been secreted in small amounts, as we also discovered a stronger band pattern of the same size as the samples generated after cell lysis (lysate). This indicates that the observed protein bands may correspond to an endogenous protein from the *P. acidilactici* cells. Our findings indicate that our particular *P. acidilactici* strain may not be capable of producing pediocin, as previously reported by Fugaban et al. [8], who suggested that pediocin production could be strain-dependent. Furthermore, it is possible that the fermentation conditions need further optimization for pediocin production.

Besides pediocin, organic acids, particularly lactic acid, are also known to play major roles in antimicrobial activity, as previously reported [33,34]. Lactic acid content in the CFS-Pa was found to increase gradually during the first 24 h of incubation to 5.02 mg/mL (Figure 1d), accompanied by a decrease in pH from 5.69 to 3.97. After 24 h, the lactic acid concentration increased slightly to a final concentration of 6.19 mg/mL, while the pH remained constant throughout the remaining incubation time (Figure 1e). The evolution of the lactic acid pattern seemed to align with the appearance of antimicrobial activity in CFS-Pa, which was observed after 12 h during the late exponential phase (Figure 1b). This suggests that lactic acid accumulation during the growth phase might have been responsible for the antimicrobial activity against the pathogens. CFS-Pa began showing its inhibitory effect against EHEC after 24 h, with a zone of inhibition reaching 6.6 ± 0.7 mm, comparable with *S. aureus* (7.7 ± 0.4 mm) and *L. monocytogenes* (6.5 ± 0.3 mm) (Figure 1b). The zone of inhibition remained stable against EHEC throughout the stationary phase, while it increased for *S. aureus* and *L. monocytogenes* to 9.4 ± 0.4 mm and 8.1 ± 0.7 mm by the end of incubation (48 h). Given that CFS-Pa collected after 48 h exhibited the highest antimicrobial activity, this sample was subjected to further characterization to better understand the metabolites involved in the antimicrobial activity of CFS-Pa.

### 3.2. Primary Metabolites Responsible for the Antimicrobial Activity of CFS-Pa

Further investigation was conducted to identify the key metabolites primarily responsible for the highest zone of inhibition exhibited by CFS-Pa collected from the 48 h culture. Among various metabolite compounds, three major metabolites commonly investigated in published studies are organic acids, bacteriocin-like compounds, and hydrogen peroxide [30,35,36,37]. Based on the measurement of total protein content and SDS-PAGE analysis (Figure 1c and Appendix A), it seems unlikely that the inhibition was caused by bacteriocin-like compounds. This is in contrast to lactic acid, which exhibits a similar pattern to the development of antimicrobial activity of the CFS-Pa (Figure 1d). Therefore, we hypothesized that organic acids play a predominant role in the antimicrobial activity. To test this hypothesis, we neutralized the pH of CFS-Pa and conducted agar well diffusion assays against *S. aureus*, EHEC, and *L. monocytogenes*. After adjusting the pH to a neutral range (6.5–7), the zone of inhibition was no longer visible, suggesting that the inhibition was exclusively due to pH or organic acids. The key inhibitory factors were also quantitatively assessed by measuring the viable cell counts of each pathogen after incubation with untreated CFS-Pa, as well as CFS-Pa following pH neutralization and treatment with pepsin and catalase (Figure 2). The results corroborated those obtained from the agar well diffusion method, showing no viable cells of any pathogens following treatment with CFS-Pa that had been pH neutralized. Similarly, no viable cells were detected when CFS-Pa was treated with either pepsin or catalase. Viable cells of approximately 9 log CFU/mL (all pathogens) were only detectable when CFS-Pa was pH neutralized. These findings further confirm that the antimicrobial properties of CFS-Pa are attributed exclusively to organic acids.

Further testing (Figure 3) indicated that organic acids in CFS worked well against both Gram-negative and Gram-positive bacteria, in comparison to lactic acid, which is only effective against Gram-positive bacteria. Kim and Kang [12] also reported similar observations regarding the antifungal activity of CFS derived from *P. acidilactici* HW01. The authors suggest that the inhibition against *Candida albicans* is exclusively associated with acid production. This is because the inhibition was reduced when the pH was adjusted to 6.12, while it remained unaffected by treatment with protease, lipase, and catalase. In that case, the lack of pediocin’s role in the inhibition is more likely because it is inherently ineffective against yeast cells [24]. Meanwhile, in the present study, the lack of bacteriocin contribution to the antimicrobial activity of CFS-Pa, even against Gram-positive bacteria, may be attributed to the absence or very low quantities of bacteriocin in the CFS-Pa. Furthermore, Castellano et al. [38] also found that an organic acids mixture (lactic acid and acetic acid, both at 2.5%) outperformed semi-purified bacteriocins combined from *L. curvatus* CRL705 and *L. sakei* CRL1862 against *L. monocytogenes* and psychrophilic microbiota on the surface of beef frankfurters.

### 3.3. Determination of Minimum Inhibitory Concentration (MIC) and Time–Kill Assay

The MIC is defined as the lowest concentration of an antimicrobial agent required to completely inhibit the growth of microorganisms [39]. In this study, we determined the MIC value of CFS-Pa exhibiting the highest antimicrobial activity, which was obtained after 48 h of incubation. The MIC values of CFS-Pa against each pathogen are detailed in Table 1. Our findings indicate that the MIC for CFS-Pa is at 10% *v*/*v* since this concentration was the minimum amount required to completely eliminate all three pathogens. The MIC values may be influenced by the pH of CFS-Pa, as demonstrated in Table 1, where an increase in dilution factor also resulted in a pH increase. For instance, CFS-Pa with a concentration of 5% *v*/*v* was no longer effective against the pathogens, and its pH was 5.16, which was higher than that of 10% *v*/*v* (pH 4.44). These findings further confirm the predominant role of pH in the antimicrobial properties of CFS-Pa.

Furthermore, we conducted a time–kill assay to evaluate the extent and rate of pathogen inhibition by CFS-Pa at its MIC (10% *v*/*v*). As depicted in Figure 4, CFS 10% *v*/*v* completely eliminated all pathogens, with EHEC exhibiting the fastest inhibition (12 h), followed by *L. monocytogenes* (18 h) and *S. aureus* (24 h). In contrast, the controls treated with MHB or MRS broth did not show inhibition at all. These results suggest that CFS-Pa at a concentration of 10% *v*/*v* exhibits bactericidal activity against the test pathogens, as it successfully reduced viable cell counts by ≥3 log CFU/mL [40].

### 3.4. Antioxidant Activity of CFS-Pa

The DPPH radical scavenging activity assay, commonly utilized for the in vitro evaluation of antioxidant capacities, was used to evaluate the antioxidant activity of CFS-Pa (T48) in this study [41,42]. The results showed that the scavenging activity was dependent on the concentration of CFS-Pa. It was found that CFS-Pa effectively scavenged a minimum of 50% of the free radicals at concentrations as low as 70% (*v*/*v*) (Figure 5). These results indicate the presence of metabolites with antioxidant properties in CFS-Pa.

Previous studies have investigated the dose and strain dependence of CFS of various LAB strains [26]. At a concentration of 10% (*v*/*v*) CFS, the CFS-PA investigated in this study shows similar if not slightly higher DPPH scavenging activity compared to the CFS of *L. casei*. It is worth noting that *L. casei* demonstrated the highest antioxidant activity among the other LAB strains tested. LAB, including *P. acidilactici*, have been reported to exhibit remarkable antioxidant activity, making them attractive for use as additives in both food and feed products. The incorporation of antioxidants in these products not only enhances their health benefits but also prolongs their shelf life, preventing the chemical (oxidization) reactions leading to unpleasant taste and/or smell [43]. The inoculation of *P. acidilactici* J17 was reported to improve the antioxidant status of alfalfa silage by enhancing the total antioxidant capacity, catalase activity, and concentrations of α-tocopherol and β-carotene [44]. A study by Incili et al. [45] demonstrated that CFS derived from *P. acidilactici* contained a wide variety of metabolites, including phenolic and flavonoid compounds, which have been found to enhance the antioxidant activity of CFS.

### 3.5. Potential Application of CFS-Pa-Loaded BC against Foodborne Pathogens

Considering the potential antimicrobial activity of CFS-Pa, we conducted additional tests to explore its potential for imparting antimicrobial properties to BC or cellulose produced by bacteria—in this case, *K. intermedius*. BC is increasingly recognized for its potential use in edible coatings and food packaging due to its biodegradable nature [46]. Moreover, BC possesses desirable properties, including food-grade quality, biodegradability, high purity, and attractive mechanical properties [22,47], making it an excellent choice for applications in food packaging and edible coatings. Our laboratory has successfully optimized various parameters crucial for enhancing BC yield [48]. However, the application of BC for edible coating and food packaging faces a challenge as BC lacks intrinsic antimicrobial properties, which are essential for extending the shelf life of food products. Therefore, the inclusion of additional substances is necessary to further prevent food deterioration and enhance stability during storage [49].

In this experiment, CFS-Pa was loaded into BC and tested against *S. aureus*, EHEC, and *L. monocytogenes* in order to investigate the potential use of CFS-Pa to create an edible food packaging material with improved antimicrobial properties. After loading CFS-Pa into BC for 8 h at 30 °C with agitation at 150 rpm, the antimicrobial activity of the loaded BC can be seen in Table 2. CFS-Pa-loaded BC was able to inhibit the growth of *S. aureus* and *L. monocytogenes* with ZOIs of 16.37 mm and 12.1 mm, respectively. Meanwhile, the lowest activity was against EHEC, where a ZOI of only 3.17 mm was observed. However, these results show that CFS-Pa-loaded BC is able to exert a wide spectrum of antimicrobial activity, comparable to other cellulose-based food packaging [50,51]. These findings suggest that the amount of antimicrobial compounds loaded into BC may have been sufficient to inhibit the tested food pathogens.

In a previous study, BC was found to exhibit selective behavior, preventing larger molecules from migrating to it, and subsequently concentrating molecules that are able to diffuse through its inner layer [28]. To further enhance the production of CFS-Pa-loaded BC, future research could involve optimizing the loading process by increasing the CFS-Pa concentration or extending the loading time. Alternatively, CFS-Pa-loaded BC could also be compared with other materials loaded with CFS-Pa to obtain the most promising food packaging [51,52]. Nevertheless, exploring the potential of CFS-Pa-loaded BC as a material for food packaging requires further study with a focus on its physicochemical properties, such as water vapor permeability and mechanical properties [51,53,54]. Finally, bactericidal activity can be assessed both in vitro and in various real meat products to further prove the efficacy of CFS-Pa-loaded BC as a food packaging [54,55,56,57].

## 4. Conclusions

In conclusion, this study explored the production and antimicrobial properties of CFS-Pa derived from *P. acidilactici.* The growth pattern of *P. acidilactici* at the end of the stationary phase showed the highest antimicrobial activity, with bacteriocin and organic acids, particularly lactic acid, possibly contributing to the observed effects. However, the SDS-PAGE analysis indicated that the proteinaceous compound in CFS-Pa may not be primarily pediocin (no band observed below 5 kDa), and further investigations pointed towards organic acids as the main contributors to the antimicrobial activity. As organic acids play a vital role, the MIC value was found at a pH of 4.44 from CFS-Pa stock at 20%. The identified MIC (10%) also managed to kill the pathogens by 24 h. Additionally, the antioxidant activity of CFS-Pa highlighted its potential applications in food preservation.

Furthermore, the incorporation of CFS-Pa into bacterial cellulose (BC) for potential use in edible food packaging demonstrated promising results, with 100% CFS-Pa-loaded BC exhibiting substantial antimicrobial activity against *S. aureus* and *L. monocytogenes*. The findings suggest that optimizing the loading process, potentially by increasing CFS-Pa concentration or extending loading time, could further enhance the antimicrobial properties of BC. This study provides valuable insights into the antimicrobial and antioxidant potential of CFS-Pa for future research and applications in the field of food preservation and packaging.

## Figures and Tables

**Figure 1 foods-13-00644-f001:**
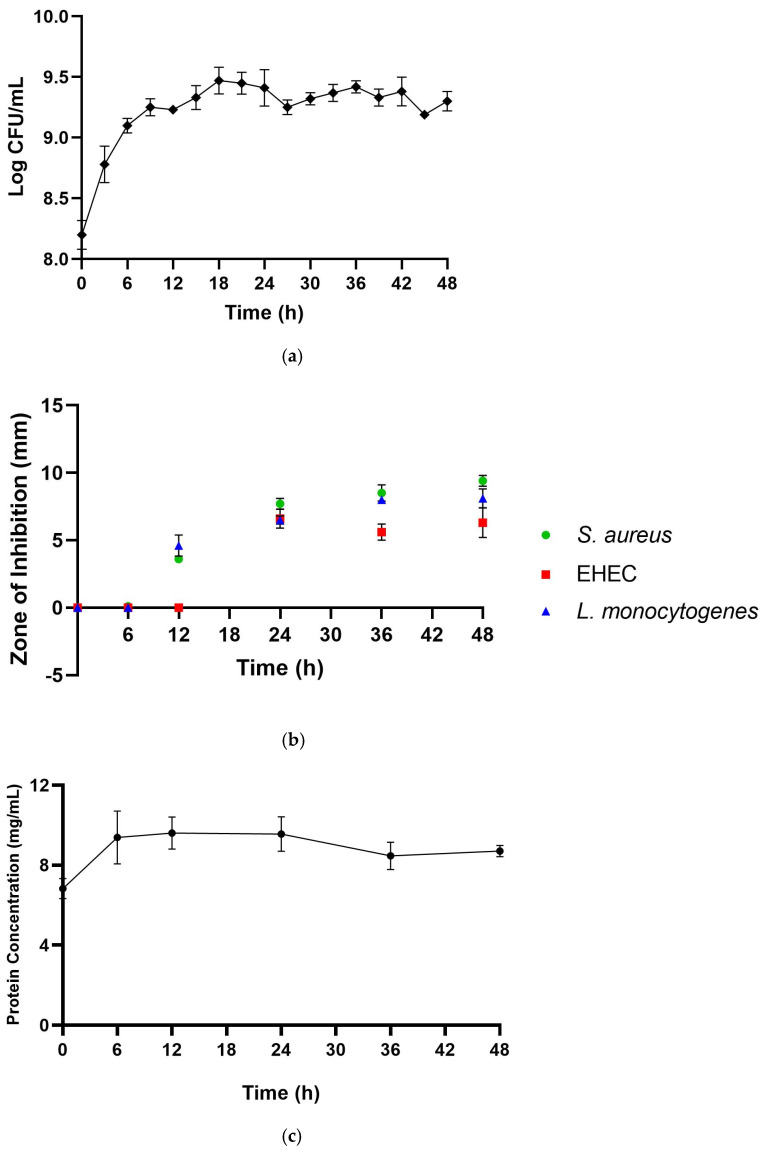
(**a**) Growth curve of *Pediococcus acidilactici* at 37 °C for 48 h. (**b**) Zones of inhibition (ZOIs) against *S. aureus* (green circle), EHEC (red square), and *L. monocytogenes* (blue triangle) by CFS harvested at the assigned time. (**c**) Protein concentration of *Pediococcus acidilactici* CFS harvested at the assigned time. (**d**) Lactic acid concentration of *Pediococcus acidilactici* CFS harvested at the assigned time. (**e**) pH value of *Pediococcus acidilactici* CFS harvested at the assigned time. Error bars represent ± standard deviation of three replicates (*n* = 3).

**Figure 2 foods-13-00644-f002:**
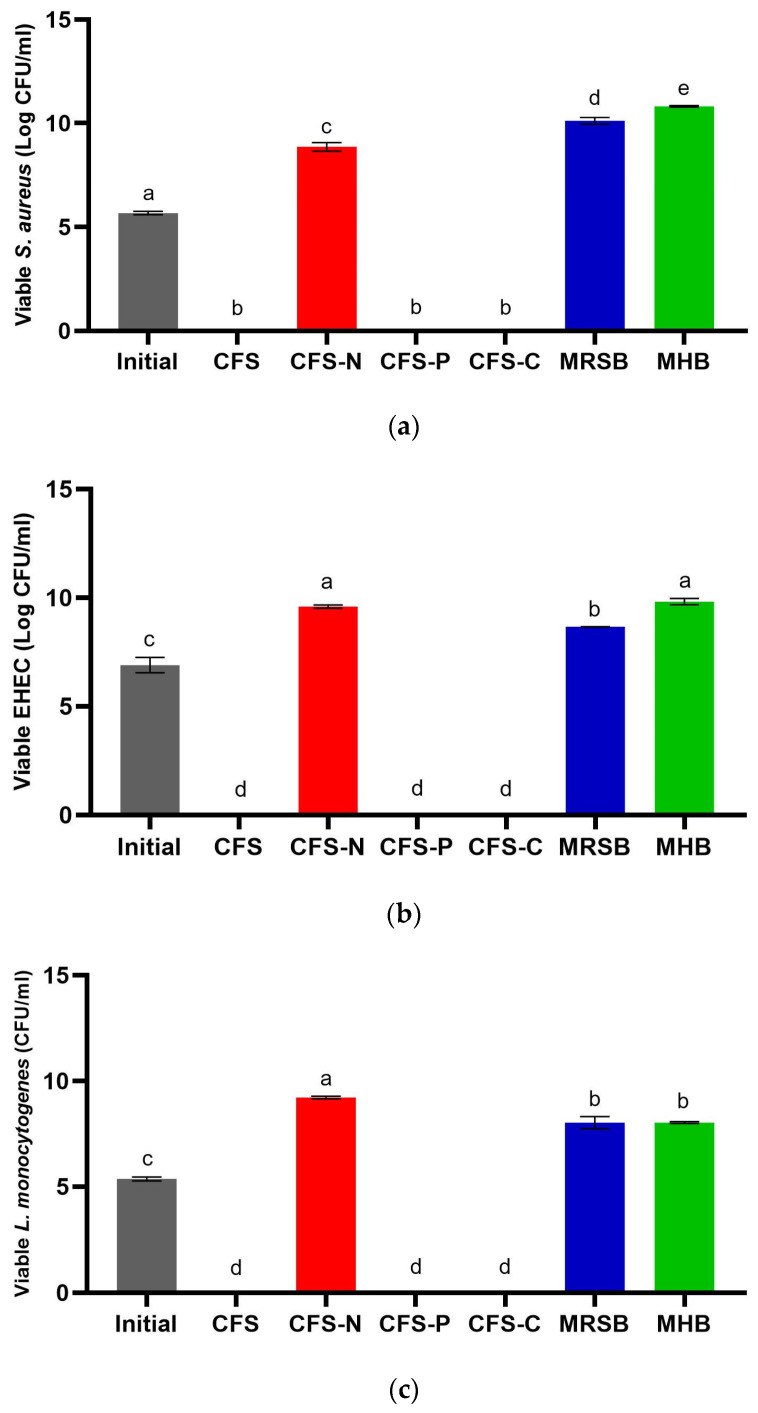
Viable cell counts of (**a**) *S. aureus*; (**b**) EHEC; and (**c**) *L. monocytogenes* after incubation with untreated CFS (CFS), neutralized CFS (CFS-N), CFS treated with pepsin (CFS-P), and CFS treated with catalase (CFS-C) for 24 h at 37 °C. Initial cell counts were performed before the incubation; MRS broth was used as the negative control; Mueller–Hinton Broth was used as the blank. Error bars represent ± standard deviation of three replicates (*n* = 3). Different superscript letters (a–e) indicate significant differences in each sample concentration. Statistical analyses were performed using ANOVA Tukey’s multiple comparison test at *p* < 0.05.

**Figure 3 foods-13-00644-f003:**
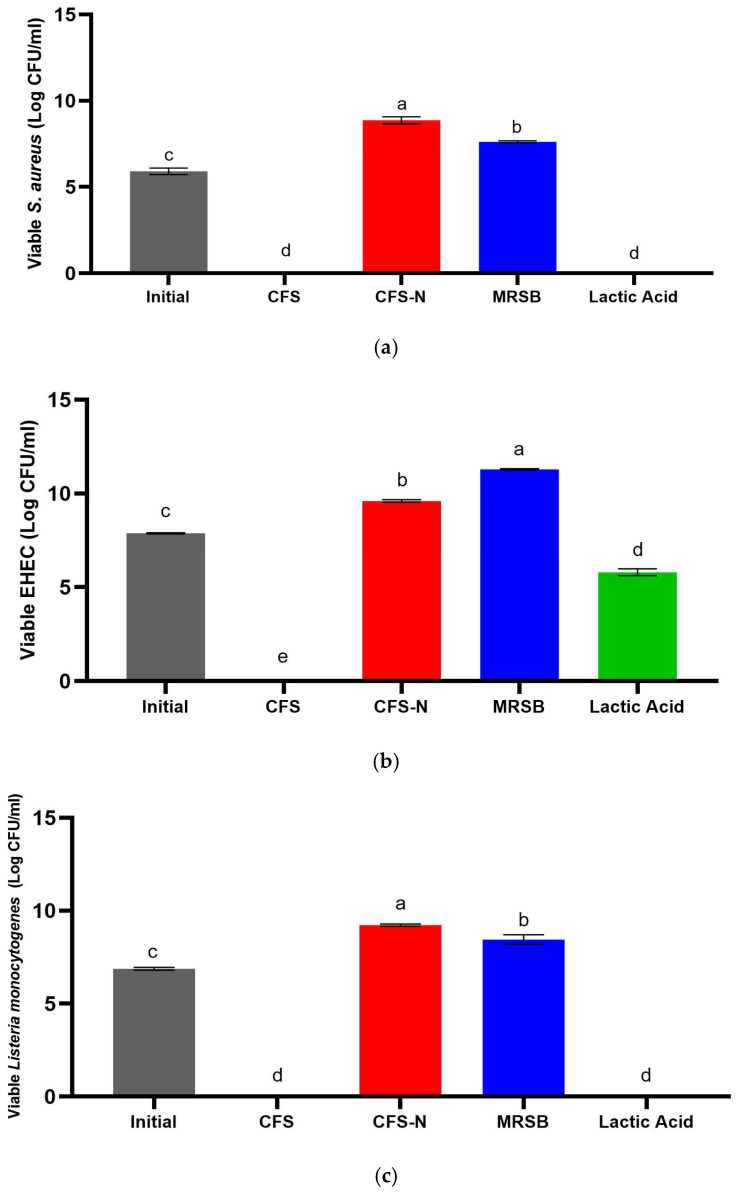
Viable cell counts of (**a**) *S. aureus*; (**b**) EHEC; and (**c**) *L. monocytogenes* after incubation with untreated CFS and neutralized CFS (CFS-N) for 24 h at 37 °C. Initial cell counts were performed before the incubation; MRS broth was used as the negative control; lactic acid (2.5 mg/mL) was used as the reference. Error bars represent ± standard deviation of three replicates (*n* = 3). Different superscript letters (a–e) indicate significant differences in each sample concentration. Statistical analyses were performed using ANOVA Tukey’s multiple comparison test at *p* < 0.05.

**Figure 4 foods-13-00644-f004:**
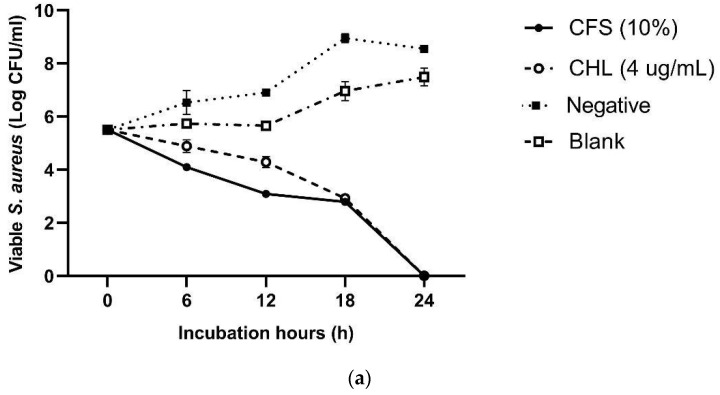
Viable cell counts of (**a**) *S. aureus*; (**b**) EHEC; and (**c**) *L. monocytogenes* after incubation with 10% of CFS for 24 h at 37 °C. Chloramphenicol (CHL) at MIC was used as the positive control; MRS broth was used as the negative control; Mueller–Hinton Broth was used as the blank. Error bars represent ± standard deviation of three replicates (*n* = 3). Statistical analyses were performed using ANOVA Tukey’s multiple comparison test at *p* < 0.05.

**Figure 5 foods-13-00644-f005:**
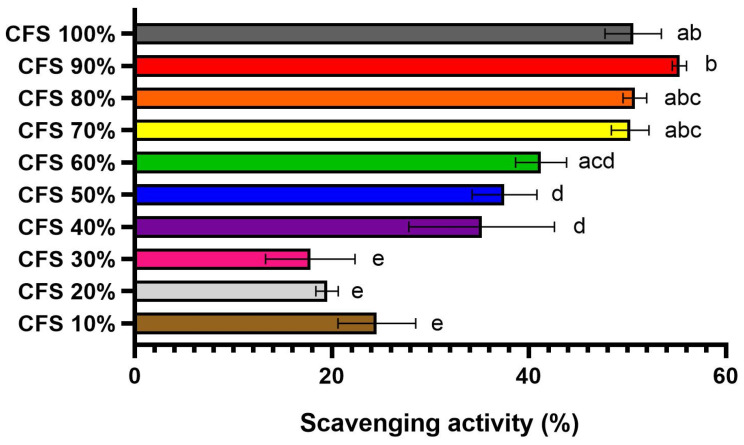
Antioxidant scavenging activity of different *Pediococcus acidilactici* CFS concentrations. Error bars represent ± standard deviation of three replicates (*n* = 3). Different superscript letters (a–e) indicate significant differences in each sample concentration. Statistical analyses were performed using ANOVA Tukey’s multiple comparison test at *p* < 0.05.

**Table 1 foods-13-00644-t001:** MIC of *Pediococcus acidilactici* CFS harvested after 48 h against *Staphylococcus aureus*, EHEC, and *Listeria monocytogenes*.

	CFS-Pa Concentration after Dilution						MRS Broth	MHB
	50%	40%	30%	20%	10%	5%		
*Staphylococcus aureus*	−	−	−	−	−	+	+	+
EHEC	−	−	−	−	−	+	+	+
*Listeria monocytogenes*	−	−	−	−	−	+	+	+
pH of CFS-Pa stock	3.75 ± 0.02	3.88 ± 0.08	4.09 ± 0.01	4.37 ± 0.02	4.44 ± 0.07	5.16 ± 0.01	5.73	7.30

+ Pathogen growth observed. − Pathogen growth not observed.

**Table 2 foods-13-00644-t002:** Antimicrobial activity of BC loaded with CFS-Pa against *S. aureus*, EHEC, and *L. monocytogenes*.

Pathogens	Zone of Inhibition (mm)
*S. aureus*	16.37 ± 1.14
*L. monocytogenes*	12.10 ± 0.78
EHEC	3.17 ± 0.92

## Data Availability

The original contributions presented in the study are included in the article/Appendix A, further inquiries can be directed to the corresponding author.

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
