# Peer review of "The Potential of Pediococcus acidilactici Cell-Free Supernatant as a Preservative in Food Packaging Materials"

_foods, 2024, doi:10.3390/foods13050644_

Round 1

Reviewer 1 Report

Comments and Suggestions for Authors

I reccommend the paper for publication after minor revision:

a) include citations/references for all metodologies, according to which standards tests were performed.

b) discusion related to antioxidant and antibacterail activity needs to be improved. Particularly, authors should include the comparison of antioxidant activity and MIC of their preservative with common synthetic preservatives for food and for other natural preservatives reported in literature. What are the benefits of preservative obtained in this study?

Comments on the Quality of English Language

Minor editing of english requested.

Reviewer 2 Report

Comments and Suggestions for Authors

The manuscript describes the antimicrobial properties of Pediococcus acidilactici Cell-free Supernatant (CFS-Pa) with a detailed analysis of the factors that ensure the effectiveness of this action to reference microorganisms: S. aureus, Enterohaemorrhagic E. coli (EHEC) and L. monocytogenes. Additionally, the antioxidant activity of CFS derived from P. acidilactici was also evaluated. The Authors included as well results of preliminary studies on the application of CFS-Pa in bacterial cellulose as an example of active packaging material – this part of the research is the least detailed in the article. Nevertheless, the manuscript meets the requirements for a research article and need minor revision.

The following items need to be corrected:

1.     A literature note (an article link) is placed in the heading of chapter 2.5.6.

2.     Poor quality of Figures 1 a-e

3.     The heading of Table 2 indicates that it presents the results of testing the antimicrobial activity of bacterial cellulose with the addition of 20% and 100% CFS-Pa, while the table contains results only for the 100% addition.

Reviewer 3 Report

Comments and Suggestions for Authors

This manuscript is an original research deals with the Potential of Pediococcus acidilactici Cell-free Supernatant as Preservative in Food Packaging. Authors have made a very good work which must be revised before farther evaluation. Please see the comments in the attached file.

Best wishes for a happy new year!

Round 2

Reviewer 3 Report

Comments and Suggestions for Authors

All requested revisions heve been made by authors.

No additional revisions required.

Accept!

Best wishes